

# A novel smart multilevel security approach for secure data outsourcing in crisis

Akram Y. Sarhan

Department of Information Technology, College of Computing and Information Technology, University of Jeddah, Jeddah, Mecca Province, Saudi Arabia

## ABSTRACT

The Interconnected Network or the Internet has revolutionized digital communications. It has expanded worldwide over the past four decades due to numerous features such as connectivity, transparency, hierarchy, and openness. Several drawbacks, including mobility, scalability, controllability, security, *etc.*, have been presented due to continuous developments. Although several network paradigms exist to address such drawbacks, many issues still persist. This research proposed a future network paradigm that addresses multilevel security shortcomings. It suggested the following: (i) a two-router network-based cyber security architecture for multilevel data sharing; (ii) using a scheduler to deal with the multilevel transmitted packets scheduling problem; (iii) five algorithms for the studied difficult problem; and (iv) providing an experimental result to show the optimal results obtained by the developed algorithms and comparing it with algorithms in the literature. The experimental result shows that the random-grouped classification with shortest scheduling algorithm (RGS) performed the best at 37.7% with a gap of 0.03. This result proves the practicality of our approach in terms of two-machine scheduling problems.

## INTRODUCTION

Anonymous data outsourcing during a crisis have become a challenge these days because of the following reasons (i) the daily amount of data produced and exchanged on the Internet; (ii) the ethical and unethical surveillance; (iii) the complexity of internetworking management; and (iv) the broken protocol designs and architectures. In fact, our world has become data-driven due to such factors as the digitization in society and economy and the advances in disruptive technologies, including unmanned aerial vehicle (UAV), artificial intelligence (AI), big data, the Internet of Things (IoT), blockchain, and robots (*Al-Gburi et al., 2022*; *Ali et al., 2022*). Another factor is the diversity of technological platforms and social services, such as YouTube, Google, Facebook, cloud computing, and mobile devices. Such services and technologies have produced enormous volumes of Internet traffic that enable information exchange and inspire the world to become data-driven (*Luo, 2022*; *Casado et al., 2006*).

Corresponding author
Akram Y. Sarhan, asarhan@uj.edu.sa

However, the endless increase in information accessibility has become a challenge thus bringing several issues to the available data, such as analysis, transmission, security, and privacy. Moreover, the imperfections of the existing Internet protocol (IP) network architecture make it difficult to address all the issues related to any data-driven model. Scalability, security, energy-saving, quality of service, and mobility are the significant issues inherited in the IP network architecture. For example, information exchange in traditional computer networks relies on the Open System Interconnection (OSI) model. The exchange of information among the OSI computing systems is separated into seven abstraction layers: application, presentation, session, transport, network, data link, and physical layers. An underlying communication routing protocols manage each layer. Thus, internetworking management is complex because each router device is responsible for routing, controlling, forwarding, and filtering packets (*Sarhan, Jemmali & Ben Hmida, 2021*). Furthermore, the IP addresses and domain name systems (DNS) are not decentralized which is a single point of failure (*Kärkkäinen, 2015*). Such a complicated, insecure traditional model needs to be secure and simplified. Several technologies have been proposed to simplify such a complex paradigm (*Sarhan, Jemmali & Ben Hmida, 2021*; *Sawalmeh & Othman, 2018*) for instance, Delay-tolerant Networking (DTN), and Software-Defined Networking (SDN).

The idea of fixing the Internet by dealing with the broken design and architecture of the current internetworking and building a new network from the scratch has launched several funded projects as cited in (*Lan et al., 2022*). The projects are as follows: the Future Internet Design (FIND), the Global Environment for Networking Innovations (GENI), the future Internet research and experimentation (FIRE), AKARI of Japan, Named Data Networking (NDN), 4WARD, MobilityFirst, ChoiceNet, FIRST, NEBULA, and the Service-Customized Networking (SCN) research projects (*Lan et al., 2022*; *Lemin, 2013*; *Zhang et al., 2010*; *Raychaudhuri, Nagaraja & Venkataramani, 2012*; *Harai, 2009*; *Brunner et al., 2010*; *Wolf et al., 2014*; *Jinho, Bongtae & Kyungpyo, 2009*; *Anderson et al., 2014*; *Liu et al., 2014*; *Greenberg et al., 2005*). Consequently, future network architecture should include several core characteristics, openness, reliability, robustness, controllability, scalability, adaptability, high performance, availability, security, credibility, manageability, highly cost-effective, and ubiquitous services, to name a few, (*Lan et al., 2022*).

Because of the emerging technologies and the broad technological and research advancements in computer networks and communications, network security, vulnerability, risks and threats are gradually expanding. Consequently, the rising number of cyber security attacks, including ransomware, denial-of-service, password, and phishing attacks, led to massive data breaches and losses in several reputable financial and industrial businesses, including government and military networks. This advancement in computer networks makes people distrust enterprises, which, in turn, leads firms to mistrust traditional tools and safeguards (*Xue, Tang & Fang, 2022*; *Fedele & Roner, 2022*; *Kärkkäinen, 2015*).

Military networks use the public network as the primary means of communication. Thus, it is targetable for several threats, including vulnerabilities and cyber security attacks. However, developing a military-based Network-enabled capability in a reasonable time is unrealistic due to the complexity of global internet governance. For example, in 2019, the US government banned Huawei—a major telecom giant company, to prevent China

from having superior control over cyberspace governance (*Kärkkäinen, 2015*; *Tang, 2020*). Hence, there is a need for a partial resolution like designing secure network architectures that can provide a timely solution in response to the urgent protection needs in such a critical armed force environment. This network should provide multilevel security, privacy protection, and delay-tolerant networking (*Kärkkäinen, 2015*).

This article proposes a two-router network-based cyber security architecture that offers private and secure multilevel data sharing. Hence, we develop a number of algorithms to achieve the goal of this article. The proposed algorithms can be applied to enhance the monitoring system developed by (*Melhim et al., 2020*; *Melhim, Jemmali & Alharbi, 2019*). On the other hand, the algorithms developed by *Jemmali (2019a)*; *Jemmali (2019b)*; *Jemmali (2022)*; *Alharbi & Jemmali (2020)*; *Jemmali (2021a)*; *Jemmali (2021b)*; *Jemmali, Otoom & Al Fayez (2020)*; *Jemmali, Melhim & Al Fayez (2022)* can be enhanced then applied to address the proposed problem. Our solution suggests enhancing the current IP network architecture by providing multilevel data security and privacy protection. Even though other issues in the existing IP network architecture are outside the scope of this research, our approach has the following pros: (i) it employs algorithmic techniques for future private networks; (ii) it provides support for anonymous communication and secure and anonymous data sharing during a crisis and in various domains like the military, pandemics, journalism, and news coverage; (iii) it presents several approximate algorithms for an NP-hard problem and uses it for secure data dissemination; (iv) it uses known and unknown algorithmic techniques such as randomization method, iterative approach and probabilistic method; (v) it presents good optimal time for the problem as it shown in the Result section.

To be more specific, let's take the example of a journalist wants to report private information about a violation anonymously during a military disaster, natural disaster, health pandemic, earthquake, flood, *etc.* suppose such confidential information demands to be communicated anonymously and promptly. In this case, there will be a need for a novel architecture that minimizes the risk of such highly confidential information breaches. Our scheme includes the following drawbacks. (1) The proposed problem is difficult; hence solving it *via* n-hopes might be complex and requires advanced algorithmic techniques of big O complexities. (2) There is a need to use a lower bound in a branch-and-bound algorithm to develop an exact solution for the problem.

The current research includes the following sections: the first discusses the related literature. The second defines the problem. The third describes the architecture and design of the proposed approach. that the fourth, introduces the proposed algorithms, reports the results, and discusses the performance measurement. The last section is summary of the article and a discussion of future work.

# LITERATURE REVIEW

## An overview of current and future network-based technologies challenges

An old or traditional computer network is a hardware-based or physical network that employs protocols based on the TCP/IP suite and requires several network devices, such as switches and routers. This interconnected system, the "Internet," has expanded worldwide since its establishment. However, its complex nature has made it undesirable due to several challenges like flexibility, security, connectivity, complexity, and bandwidth. On the other hand, computer networks, nowadays, attract significant attention because of the wide adoption of new features and technologies like automaticity, AI, cloud computing, machine learning, SDN, and IoT.

SDN, on the other hand, has been considered by many as one of the possible future network paradigms due to its benefits in strengthening network architecture, reducing operational costs, and supporting the addition of new applications and functions. It is a software-based architecture that simplifies and improves network control by isolating the control from the forwarding plane, thus making it practical to add new network functions or protocols. However, SDN possesses security concerns and other issues (*Benzekki, El Fergougui & Elbelrhiti Elalaoui, 2016*; *Shin et al., 2016*; *Duan, Yan & Vasilakos, 2012*; *Lan et al., 2022*). Therefore, SDN is used by (*Shin et al., 2016*) to enhance network security and information security processes. Since SDN is considered the foundational building block of Intent-Based Networking (IBN), it functions to address SDN's shortfall. For example, IBN is proposed to deal with system requirements without going into detail. In IBN, the system behaviors are chosen by rules which are considered a kind of policy. The current focus on IBN is still inside academia. However, there is an expectation for future adoption of IBN by leading cloud vendors due to advancements in AI, specifically in natural processing language (NLP) (*Rafiq, Afaq & Song, 2020*; *Zeydan & Turk, 2020*).

In addition, the fifth-generation technology (5G) wireless network aims to address 4G challenges such as; data rate, spectral and energy efficiency, capacity, and Quality of Service (*Gupta & Jha, 2015*). However, 5G has issues like authentication and data security (*Sivasubramanian, Shastry & Hong, 2022*).

In another vein, Cloud computing is a collection of shared resources that includes computer networks, storage, services, and servers. The three standard cloud computing service paradigms are Infrastructure as a Service (IaaS), Platform as a Service (PaaS), and Software as a Service (SaaS), which can be managed and hosted through an independent third-party provider and accessed through the Internet. Cloud networking is also concerned with methods to access cloud applications. It can be accessed and operated through personal and BYOD devices like desktop computers and other "BYOD" interfaces or personal devices that can access the Internet, like laptops, smart phones, and personal computers (*Hong et al., 2019*). Besides maintenance and cost saving, efficiency, and workload flexibility, the cloud enables organizations from various domains to share and exchange data to perform analysis and extract patterns that can find solutions for multiple problems (*Sarhan & Carr, 2017*). Accessing cloud computing can be through one or more efficient deployment

models: private, public, hybrid, or multi-cloud. Cloud technology has many challenges and drawbacks which are presented as follows: (i) centralization (external third party manages data computation and storage); (ii) high latency; (iii) interoperability; (v) data security, privacy, and management. Unlike the hybrid cloud, which relies on multiple deployment modes, the multi-cloud technology model utilizes various cloud services simultaneously from multiple cloud service providers. It addresses challenges presented in other cloud models (*Hong et al., 2019*).

The age of the Internet of Things (IoT) has made a massive number of devices connect to the cloud thus creating several issues related to cloud centralization. Edge computing can overcome problems related to the centralization nature of the cloud, such as network bandwidth and bandwidth cost, latency, IoT battery life constraints, and privacy because the data processing and computation happen at the network's edge (*Shi et al., 2016*). However, the drawback of Edge Computing is security (*Jiang et al., 2015*; *Hossain, Fotouhi & Hasan, 2015*; *Yang et al., 2017*). Cloud repatriation is a new concept proposed to overcome security issues in the cloud. Compared to edge computing, cloud repatriation eliminates issues created by the public cloud, like operation cost, performance, control, and security (*Shin et al., 2016*; *Hintemann, 2020*). However, cloud repatriation is still under consideration.

## Network security challenges

The current network security challenges have resulted from the poor, insecure traditional and complex architecture of the Internet. As a result, several approaches have been presented to provide architectures for such issues. For instance, *Casado et al. (2006)* published a scheme for protecting network architecture. The so-called "SANE" scheme provides strict security policy control for private networks. It prevents illegal interaction and requires the source and destination to be declared.

XIA aims to provide a single network infrastructure that controls the network and removes communication obstacles between the end users and the network infrastructure. It uses an application program interface (API) for port-to-port communication. The security mechanism in XIA, the so-called "intrinsic security mechanism," is implemented through a unified network infrastructure in which each user has security identification applied to credit management. Furthermore, the security control in XIA increases from single packet forwarding to interoperation among the network components (*Berman et al., 2014*).

NEBULA aims to provide built-in security and adaptable central network architecture that uses cloud computing data centers for storing and computing data. As a result, it can solve cloud computing which causes the emergence of security threats (*Liu et al., 2014*).

FIRE aims to provide network architecture and protocols for future intelligent Internet that address security, complexity, scalability, and mobility issues. Its interconnected smart networks should support intelligent transportation, medical, and social life (*Gavras et al., 2007*). MobilityFirst focuses on providing mobile services architecture to design future Internet that is based on mobile devices. Its main intention is to address security, privacy, availability, manageability, and tolerance (*Naylor et al., 2014*).

## Packets security and privacy challenges

*Alzahrani & Chaudhry (2022)* proposed an SDN securing source routing forwarding scheme that provides packet protection by using a cryptographic authenticator to authorize SDN switches and impose a selected routing path. Their approach uses identity-based encryption (IBE), considers single and multipath transmissions, and allows the receiver to authenticate the envisioned path of the forwarded packets. The drawback of their scheme, however, is the security-performance overhead.

*Zeng, Zhang & Xia (2022)* proposed a blockchain-based SDN network architecture for the security of routing among numerous hosts. However, their scheme relies on the reputation concept for routing reliability which cannot be absolute. *Legner et al. (2020)*, on the other hand, proposed EPIC protocols to secure the inter-domain paths at the Internet inter-autonomous communication system levels. They used symmetric key encryption for packet authentication between the sender and receiver at the network layers. In addition, *Singh et al. (2021)* proposed a secure scheme that controls the traffic flow by integrating blockchain with switches. Their solution uses deep learning and a zero-knowledge proof technique to verify the registered switches in the network.

## Multilevel data sharing approaches

*Zaghloul, Zhou & Ren (2020)* suggested a cloud-secure and efficient multilevel data outsourcing solution. The scheme divides the outsourced data into different parts to share it according to (i) the user's authorized privileges and (ii) the level of confidentiality of the outsourced data.

*Sarhan & Lilien (2014)* also proposed a novel multilevel data-outsourcing approach to protect outsourced data in the cloud. The scheme uses Secure Multi-Party Computation, cipher text policy attribute-based encryption (CP-ABE), and active bundles to encapsulate the data with its access policy within a virtual machine. Besides using CP-ABE that permits multilevel data access, the schemes encapsulate two attributes: location and time to strengthen the protective layer for the outsourced data in the cloud (*Sarhan & Carr, 2017*; *Sarhan, Jemmali & Ben Hmida, 2021*; *Sarhan, 2017*).

## Our approach *versus* the other two router' approaches

Alquhayz et al. approach uses a scheduler for packet prioritizing based on data multilevel security constraints. They proposed several heuristics and performed simulation experimentation using a static window pass based on a single router (*Jemmali & Alquhayz, 2020a*; *Jemmali & Alquhayz, 2020b*; *Alquhayz & Jemmali, 2021*). In another work (*Jemmali & Alquhayz, 2020a*; *Jemmali & Alquhayz, 2020b*), they used identical routers for scheduling problems.

On the other hand, Sarhan et al.'s packet multilevel security scheme uses a constraint-based packet categorization and dissemination and two routers. In addition, the authors suggested several scheduling algorithms to minimize the transmission time (*Sarhan, Jemmali & Ben Hmida, 2021*; *Sarhan & Jemmal, 2023*). This research uses different heuristics to deal with the proposed problem.

| Table 1 System notation summary. | |
|---|---|
| **Symbol** | **Description** |
| R1 | Router 1 |
| R2 | Router 2 |
| $Ct_j^1$ | the cumulative transmission time when the packet $Pt_j$ is assigned to the router $R1$ |
| $Ct_j^2$ | the cumulative transmission time when the packet $j$ is assigned to the router $R2$ |
| $Pt_j$ | set of packets |
| $n$ | Number of packets |
| $T_m$ | maximum completion time on routers |
| $Cg_i$ | categories with $i$ |
| $n_{Cg}$ | Fixed number of categories |

## PROBLEM FORMULATION

The current research focuses on the problem of protecting multilevel packets by controlling the packets' path and transmission time, which is essential in a private network. We assume that the network packets are classified into several category levels so that packets belonging to the identical classification level are prohibited from being transmitted at the same time over the two routers. This problem is a difficult problem that we handle using approximate solutions. Furthermore, we impose a security constraint that prevents two packets originating from the same confidential level from being transited simultaneously to maximize the level of outsourced data protection and minimize the chances of data leaks.

The objective of this article is to create numerous near-optimal solutions for the studied problem. We refer to $Pt$ as a group of packets and mark $n$ as their number. We denote $R1$ for router1 and denote $R2$ for router 2. When packet $Pt_j$ is sent to router $R1$, the cumulative transmission time is denoted as $Ct_j^1$ and when packet $Pt_j$ is assigned to the router $R2$, the cumulative transmission time is denoted as $Ct_j^2$. We denote $t_j$ for packet $Pt_j$ estimated transmission time. See Table 1 for more details. We denote $T_1$ and $T_2$ for the total time of transmission on $R1$ and $R2$, and $T_m$ for the routers' maximum time, so $T_m = \max(T_1, T_2)$. $Cg_i$ denotes the categories such that $i = 1, \ldots, n_{Cg}$ and $n_{Cg}$ is the number of categories fixed by the administrator. The objective is to minimize $T_m$.

### Proposition

Two routers scheduling a problem based on a multilevel security is a difficult problem because the minimization of the total time of transmission using the 2-router problem is the reduction 2-parallel machines NP-Hard problem (*Sarhan, Jemmali & Ben Hmida, 2021*; *Garey & Johnson, 1979*). This reduction is because the two routers correspond to the two machines, and the scheduling of packets corresponds to the scheduling of jobs.

## ARCHITECTURE AND DESIGN

This section provides the proposed solution architecture details and its objective design.

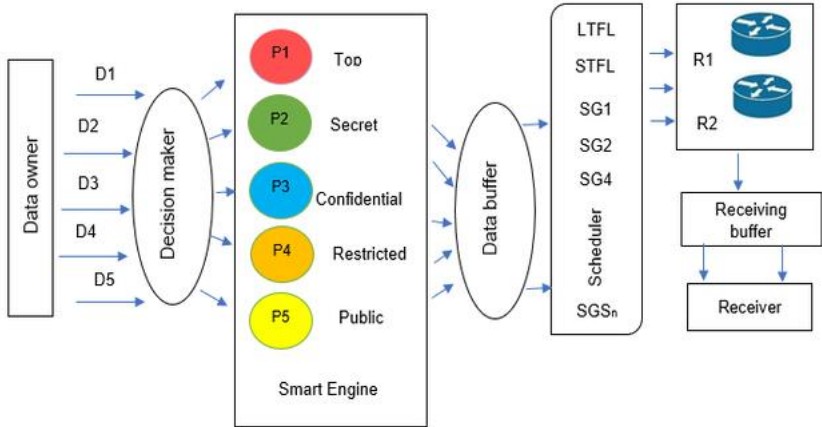

**Figure 1** **Novel architecture with two routers.**

## System model

This sub-section discusses the components of the proposed two-router network-based cyber security architecture. In Fig. 1, the constructed architecture assumes the transmissions of packets using two routers. The processes are as follows: first, the data owner and decision maker (*e.g.*, machine learning engine or agent) categorize the data into several classified levels. Next, the categorized data will be waited in a buffer and processed by a scheduler *via* a selected algorithm from a pool, which intelligently controls the packet transmitted to the two routers. The scheme is composed of several components which are described as follows:

(1) Data owner: this component manages the transmitted data's classified level, specifies the files to be sent to the key decision maker, and fixes the data level categories.

(2) Decision maker: this component represents a key decision maker, or security policy maker, who categorizes the transmitted data and their level of importance.

(3) Smart engine: this component represents a software agent that manages data transmission to the data buffer. It controls this component, administers the data and its transmission, and links the sent files with their categories after being classified.

(4) Data collection engine: this component groups all files for the transmitted data.

(5) Data buffer engine: this component collects, verifies, and links the sent files within a category.

(6) Scheduler: this component is essential for solving a scheduling problem related to data disclosure through two routers. The scheduler provides several algorithms and selects the best one to solve a particular scheduling problem. It receives the files sent from the Data buffer engine.

(7) Receiving buffer: this component stores or groups the transmitted files in the 'Receiver buffer.'

(8) Routers: this component represents the two routers.

(9) Receiver: this component represents a user expecting to receive confidential information.

## Design objective

To provide a secure future network architecture that offers multilevel data sharing for future deployment of private networks in a critical environment. Our design has the following objectives:

(i) Multilevel data access policy: designing a multilevel security policy for data classification and transmission. Note that a constraint governs the security policy and the transmitted packets.

(ii) Scheduler: designing a scheduler capable of selecting the best packet scheduling algorithm from a pool. The scheduler is assumed to perform its calculations autonomously.

(iii) Two routers: designing a network paradigm so that the classified packets do not significantly impact the transmission time.

# PROPOSED ALGORITHMS

This section presents five newly designed algorithms that solve the studied problem (*Sarhan, Jemmali & Ben Hmida, 2021*; *Sarhan & Jemmal, 2023*) in a remarkable time. The proposed algorithms use different techniques to enhance the transmitted time and reduce the algorithmic complexity. The first discussed algorithm is the "longest transmission time with excluding the first and last packet" (LTFL). In LTFL, packets are arranged in a descending sequence to exclude the longest and shortest packets with estimated transmission time and then to schedule all remaining packets. Second, we describe "the shortest transmission time with excluding the first and the last packet" (STFL) algorithm. In STFL, we arrange packets in an ascending sequence to exclude the longest and shortest packets with estimated transmission time, so the obtained schedule will be used to calculate the result. Third, we discuss the Shortest-Grouped classification (SG) algorithm. In SG, we arrange packets in an ascending sequence based on their estimated transmission time, and then we divide them into groups to schedule them through several variants to find a better solution than others. After that, we talk about the Random-Grouped classification with longest scheduling (RGL) algorithm, and finally, we present the Random-Grouped classification with shortest scheduling (RGS) algorithm. Unlike SG, In RGL and RGS sorting, the packets are delayed until the end to get the best solution.

## Longest transmission time with excluding the first and the last packet (LTFL)

In LTFL, the processes are as follows: first, we store packets in a decreasing order based on their approximate transmission time descending sequence. Second, we exclude the longest and the shortest packets. Third, we schedule the n-2 remaining packets on the faster or lowest completed-time router. Finally, we schedule the two retained packets. Note that *Dsc()* denotes the function that sorts all packets according to their estimated transmission time descending sequence, and Schd(L) denotes the function that schedules a list L on the two routers.

**Algorithm 1: Algorithm LTFL**

| | |
|---|---|
| 1 | Call Dsc() |
| 2 | L1 = the first packet |
| 3 | L2 = the last packet |
| 4 | L = all packets excluding L1 and L2 |
| 5 | Call Schd(L) |
| 6 | L3= = {L1, L2} |
| 7 | Call Schd(L3) |
| 8 | Calculate $T_m$ |
| 9 | Return $T_m$ |

**Algorithm 2: Algorithm STFL**

| | |
|---|---|
| 1 | Call Isc() |
| 2 | L1 = the first packet |
| 3 | L2 = the last packet |
| 4 | L = all packets excluding L1 and L2 |
| 5 | Call Schd(L) |
| 6 | L3= = {L1, L2} |
| 7 | Call Schd(L3) |
| 8 | Calculate $T_m$ |
| 9 | Return $T_m$ |

***Shortest transmission time with excluding the first and the last packet (STFL)***

In STFL, the processes are as follows: first, we sort the packets according to the increasing order of their estimated transmission time. Second, we exclude the longest and the shortest packets. Next, we schedule the n-2 remaining packets on the router with the minimum completion time, and finally, we schedule the two retained packets. Note that *Isc()* denotes the function that sorts all packets according to the increasing order of their estimated transmission time.

***Shortest-grouped classification algorithms (SG)***

In SG, the processes are as follows: first, we sort the packets corresponding to the estimated transmission time ascending sequence. Second, we divide the sorted packets into three groups so that each group is composed of $\frac{n}{3}$ packets. The first group $G1$ contains the first $\frac{n}{3}$ packets, the second group $G2$ contains the second $\frac{n}{3}$ packets, and the last group, $G3$ contains the remaining packets. Third, we adopt four variants to schedule the packets in these groups. We denote $SG_1$ for the first variant. The schedule of $SG_1$ packets are applied as follows: we schedule packets of $G2$, next packets of $G1$, and finally packets of $G3$. This variant is denoted by $SG_1$. We denote $SG_2$ for the second variant. The schedule of $SG_2$ packets are applied as follows: we schedule packets of $G2$, next packets of $G3$, and finally packets of $G1$. We denote $SG_3$ for the third variant. The schedule of $SG_3$ packets are applied as follows: we schedule packets of $G3$, next packets of $G1$, and finally packets of $G2$. We denote $SG_4$ for the fourth variant. The schedule of $SG_4$ packets are applied as follows: we schedule packets of $G1$, next packets of $G3$, and finally packets of $G2$.

Hereafter, we denote *Gprd()* for the function that returned the three lists related to *G*1, *G*2, and *G*3. These lists will be denoted by SG1, SG2, and SG3 respectively.

**Algorithm 3**: Algorithm $S_{G1}$

| 1 | Call Isc() |
|---|---|
| 2 | Call Gprd() |
| 3 | Call Schd (SG2) |
| 4 | Call Schd (SG1) |
| 5 | Call Schd (SG3) |
| 6 | Calculate $T_m$ |
| 7 | Return $T_m$ |

**Algorithm 4**: Algorithm $S_{G2}$

| 1 | Call Isc() |
|---|---|
| 2 | Call Gprd() |
| 3 | Call Schd (SG2) |
| 4 | Call Schd (SG3) |
| 5 | Call Schd (SG1) |
| 6 | Calculate $T_m$ |
| 7 | Return $T_m$ |

**Algorithm 5**: Algorithm $S_{G3}$

| 1 | Call Isc() |
|---|---|
| 2 | Call Gprd() |
| 3 | Call Schd (SG3) |
| 4 | Call Schd (SG1) |
| 5 | Call Schd (SG2) |
| 6 | Calculate $T_m$ |
| 7 | Return $T_m$ |

## Random-grouped classification with longest scheduling algorithms (RGL)

As detailed in the above subsection, we first divided the packets (without any sorting) into three groups. Then, we applied the same four variants described above to schedule the packets in these groups. Finally, the packets were sorted in each group according to their estimated transmission time in a decreasing order. Note that the first, second, third, and fourth variants are denoted as $RGL_1$, $RGL_2$, $RGL_3$, and $RGL_4$, respectively.

**Algorithm 6: Algorithm $S_{G4}$**

| | |
|---|---|
| 1 | Call Isc() |
| 2 | Call Gprd() |
| 3 | Call Schd (SG1) |
| 4 | Call Schd (SG3) |
| 5 | Call Schd (SG2) |
| 6 | Calculate $T_m$ |
| 7 | Return $T_m$ |

**Algorithm 7: Algorithm $RGL_1$**

| | |
|---|---|
| 1 | Call Gprd() |
| 2 | Call Dsc(GL2) |
| 3 | Call Schd (GL2) |
| 4 | Call Dsc(GL3) |
| 5 | Call Schd (GL2) |
| 6 | Call Dsc(GL1) |
| 7 | Call Schd (GL1) |
| 8 | Calculate $T_m$ |
| 9 | Return $T_m$ |

## Random-grouped classification with shortest scheduling algorithms (RGS)

As detailed in the above subsection, we divided the packets (without any sorting) into three groups. Then, we applied the same four variants described above to schedule the packets in these groups. Finally, the packets were sorted in each group according to their estimated transmission time in an increasing order. Note that the first, second, third, and fourth variants are denoted by $RGS_1$, $RGS_2$, $RGS_3$, and $RGS_4$, respectively.

**Algorithm 8: Algorithm $RGS_1$**

| | |
|---|---|
| 1 | Call Gprd() |
| 2 | Call Isc (GL2) |
| 3 | Call Schd (GL2) |
| 4 | Call Isc (GL3) |
| 5 | Call Schd (GL3) |
| 6 | Call Isc (GL1) |
| 7 | Call Schd (GL1) |
| 8 | Calculate $T_m$ |
| 9 | Return $T_m$ |

## Experimental setup

This section describes the proposed algorithms' experimental results, the variables used, and the simulation environment to measure the performance.

We used C++ to prototype the proposed algorithms. The computing environment included one gigahertz on an Intel CPU and eight gigabytes of RAM. We produced 300 instances defined as follows: $n = 15, 25, 35, 45, 55$ and $n_{Cg} = 2, 3, 5$ to measure the created algorithms' performance measurements. We used a random function called a uniform distribution to generate the estimated transmission time. We denote U[.] for the uniform distribution function.

In this article, we adopted two classes. Class 1 corresponds to the estimated transmission time generated as $U[1-50]$. Class 2 corresponds to the estimated transmission time generated as $U[1-100]$. For each pair $(n, n_{Cg})$ and for each class, we produce ten instances; thus, we calculate the total number of instances as follows: $5 \times 3 \times 2 \times 10 = 300$ instances. Furthermore, we used three variables to evaluate the created algorithms' performance time. The descriptions of the variables are as follows; (i) variable *Prc* which indicates the total instances percentage in case a given algorithm is the same as the best; (ii) variable *Dv* which shows the gap between a candidate algorithm value, say "$x$" and the best-obtained one value say "$y$". Indeed, $Dv = \frac{x-y}{y}$. (iii) variable *Tm* represents the algorithm's average time in seconds. Note that the mathematical symbol "-" indicates that the time is lower than 0.001 s.

## RESULTS & DISCUSSION

The current section describes the experimental findings of the created algorithms and compares the results with those presented in works of (*Sarhan, Jemmali & Ben Hmida, 2021*; *Sarhan & Jemmal, 2023*) to find the best algorithm. A summary of the core experimental variables and performance results is in Tables 2–6. Table 2 presents an overview of the performance of the proposed algorithms in this article. Our observation is that the best algorithm is the Random-Grouped Classification with Shortest Scheduling Algorithms (*RGS*) since its first and fourth variants ($RGS_1$ and $RGS_4$) achieved the best results recording 37.7%, an average gap of 0.03, and an estimated transmission time of 0.001 s. The algorithm based on the grouping method gives better results because there is a classification of the packets into several groups, especially when we start with the increasing order of the packets.

We also noticed that the worst algorithm is *STFL,* rating 18.3%, a gap of 0.04, and an estimated transmission time of 0.001, and the lowest average gap algorithm is $SG_2$. The STFL algorithm is a dispatching rule method, which gives priority to the packet with the shortest total transmission time. This priority makes the packet with the maximum transmission time scheduled the last. This makes it a problem to find a long packet to be transmitted in a router that has the minimum total transmission time.

Table 3 compares the variations of the average gap of all proposed algorithms when $n$ changes. We observe that both algorithms' *STFL*, and $SG_1$ average gap variations are the same regardless of the number of packets $n$. Moreover, algorithm $SG_2$ has the lowest average gap variation of 0.01 in two scenarios when $n = 45$ and $n = 55$. For this algorithm, it is clear that the average gap decreases when the number of packets increases. This means that the problem becomes simpler, and there are many packets, reflecting that more choices can be derived for the scheduling.

**Table 2  Overview of the performance of the proposed algorithms.**

|      | LTFL  | STFL  | $SG_1$ | $SG_2$ | $SG_3$ | $SG_4$ | $SGL_1$ | $SGL_2$ | $SGL_3$ | $SGL_4$ | $RGS_1$ | $RGS_2$ | $RGS_3$ | $RGS_4$ |
|------|-------|-------|--------|--------|--------|--------|---------|---------|---------|---------|---------|---------|---------|---------|
| $Prc$ | 30.3% | 18.3% | 24.0%  | 33.0%  | 35.0%  | 23.7%  | 20.0%   | 22.7%   | 23.3%   | 22.0%   | 37.7%   | 37.0%   | 34.3%   | 37.7%   |
| $dv$  | 0.05  | 0.04  | 0.04   | 0.02   | 0.03   | 0.04   | 0.05    | 0.04    | 0.04    | 0.04    | 0.03    | 0.03    | 0.04    | 0.03    |
| $Tm$  | 0.002 | 0.001 | 0.002  | 0.001  | 0.001  | 0.002  | 0.002   | 0.001   | 0.001   | 0.002   | 0.001   | 0.001   | 0.001   | 0.001   |

**Table 3  The average gap variation for all proposed algorithms when $n$ changes.**

| $n$ | LTFL | STFL | $SG_1$ | $SG_2$ | $SG_3$ | $SG_4$ | $SGL_1$ | $SGL_2$ | $SGL_3$ | $SGL_4$ | $RGS_1$ | $RGS_2$ | $RGS_3$ | $RGS_4$ |
|-----|------|------|--------|--------|--------|--------|---------|---------|---------|---------|---------|---------|---------|---------|
| 15  | 0.07 | 0.05 | 0.05   | 0.04   | 0.04   | 0.06   | 0.05    | 0.06    | 0.05    | 0.06    | 0.03    | 0.03    | 0.03    | 0.03    |
| 25  | 0.05 | 0.05 | 0.05   | 0.03   | 0.03   | 0.04   | 0.06    | 0.05    | 0.04    | 0.05    | 0.04    | 0.04    | 0.05    | 0.04    |
| 35  | 0.05 | 0.04 | 0.04   | 0.02   | 0.03   | 0.03   | 0.04    | 0.04    | 0.04    | 0.03    | 0.03    | 0.04    | 0.03    | 0.03    |
| 45  | 0.04 | 0.03 | 0.03   | 0.01   | 0.02   | 0.03   | 0.03    | 0.04    | 0.03    | 0.03    | 0.03    | 0.03    | 0.03    | 0.02    |
| 55  | 0.04 | 0.03 | 0.03   | 0.01   | 0.02   | 0.02   | 0.04    | 0.04    | 0.04    | 0.03    | 0.03    | 0.03    | 0.04    | 0.03    |

**Table 4  The average gap variation for all proposed algorithms when $n_{cg}$ changes.**

| $n_{Cg}$ | LTFL | STFL | $SG_1$ | $SG_2$ | $SG_3$ | $SG_4$ | $SGL_1$ | $SGL_2$ | $SGL_3$ | $SGL_4$ | $RGS_1$ | $RGS_2$ | $RGS_3$ | $RGS_4$ |
|----------|------|------|--------|--------|--------|--------|---------|---------|---------|---------|---------|---------|---------|---------|
| 2 | 0.01 | 0.01 | 0.01 | 0.01 | 0.01 | 0.01 | 0.01 | 0.02 | 0.01 | 0.01 | 0.01 | 0.01 | 0.01 | 0.01 |
| 3 | 0.07 | 0.06 | 0.05 | 0.04 | 0.05 | 0.06 | 0.07 | 0.06 | 0.06 | 0.06 | 0.05 | 0.05 | 0.06 | 0.05 |
| 5 | 0.07 | 0.05 | 0.06 | 0.02 | 0.03 | 0.04 | 0.06 | 0.06 | 0.05 | 0.05 | 0.04 | 0.04 | 0.03 | 0.03 |

On the other hand, the average gap variation for algorithm $LTFL$ increases when $n$ is low. For example, the highest average gap of 0.07 appears with algorithm $LTFL$ when $n = 15$.

Table 4, on the other hand, shows that all algorithms have the same average gap variation of 0.01 when $n_{Cg} = 2$ except algorithm $SGL_2$ which is equal to 0.02. The $SGL$ variant consists of the scheduled packets of $G2$, then the scheduled one of $G3$, and finally the scheduled packets of $G1$. Therefore, starting with the packets of $G2$ and ending with packets of $G1$ don't give good results.

Furthermore, algorithms $SGL_3$ and $SGL_4$ have the sameaverage gap variation regardless of the changes in the number of categories $n_{Cg}$.

Table 5 presents the time variation for all proposed algorithms when $n$ changes. We observe that the time variation for the algorithm $SGL_2$ is not influenced by the changes in the number of packets $n$ since it equals 0.001. This variant of $SGL$ consists of the scheduled packets of $G2$, next packets of $G3$, and finally packets of $G1$. Starting with the packets of $G2$ and ending with packets of $G1$ give special results compared with other algorithms.

One can also observe that the time variation for algorithm $LTFL$ gradually decreases when the number of packets $n$ increases, and the time variation for both algorithms $SGL_2$ and $SG_2$ are alike or do not change when the number of packets $n$ changes.

Table 6 displays the proposed algorithms' variation of time when $n_{Cg}$ changes. It is observable that both algorithms $SGS_2$ and $SG_4$ time variation is not influenced by the changes in the number of categories $n_{Cg}$ variation since $SGS_2 = 0.001$ and $SG_4 = 0.002$ regardless of the value of $n_{Cg}$. This reflects the complexity of algorithms $SGS_2$ and $SG_4$.

**Table 5** The time variation for all proposed algorithms when *n* changes.

| n | LTFL | STFL | $SG_1$ | $SG_2$ | $SG_3$ | $SG_4$ | $SGL_1$ | $SGL_2$ | $SGL_3$ | $SGL_4$ | $RGS_1$ | $RGS_2$ | $RGS_3$ | $RGS_4$ |
|---|------|------|------|------|------|------|------|------|------|------|------|------|------|------|
| 15 | 0.003 | 0.001 | 0.001 | 0.001 | 0.001 | 0.001 | 0.002 | 0.001 | 0.001 | 0.002 | 0.001 | 0.001 | 0.001 | 0.001 |
| 25 | 0.002 | 0.002 | 0.001 | 0.001 | 0.002 | 0.002 | 0.001 | 0.001 | 0.001 | 0.002 | 0.002 | 0.003 | 0.001 | 0.001 |
| 35 | 0.002 | 0.001 | 0.002 | 0.001 | 0.002 | 0.002 | 0.002 | 0.001 | 0.002 | 0.002 | 0.001 | 0.000 | 0.002 | 0.001 |
| 45 | 0.002 | 0.002 | 0.002 | 0.001 | 0.001 | 0.002 | 0.001 | 0.001 | 0.002 | 0.002 | 0.001 | 0.001 | 0.002 | 0.002 |
| 55 | 0.001 | 0.001 | 0.003 | 0.001 | 0.001 | 0.002 | 0.002 | 0.001 | 0.002 | 0.002 | 0.002 | 0.001 | 0.001 | 0.001 |

**Table 6** The time variation for all proposed algorithms when ncg changes.

| $n_{Cg}$ | LTFL | STFL | $SG_1$ | $SG_2$ | $SG_3$ | $SG_4$ | $SGL_1$ | $SGL_2$ | $SGL_3$ | $SGL_4$ | $RGS_1$ | $RGS_2$ | $RGS_3$ | $RGS_4$ |
|---|------|------|------|------|------|------|------|------|------|------|------|------|------|------|
| 2 | 0.001 | 0.002 | 0.002 | 0.001 | 0.001 | 0.002 | 0.002 | 0.001 | 0.002 | 0.002 | 0.001 | 0.001 | 0.002 | 0.001 |
| 3 | 0.002 | 0.002 | 0.001 | 0.002 | 0.002 | 0.002 | 0.001 | 0.002 | 0.002 | 0.002 | 0.002 | 0.001 | 0.001 | 0.002 |
| 5 | 0.003 | 0.001 | 0.002 | 0.001 | 0.001 | 0.002 | 0.002 | 0.001 | 0.001 | 0.001 | 0.001 | 0.001 | 0.001 | 0.001 |

One can also observe that the time variation for algorithm *LTFL* gradually increases when the value of $n_{Cg}$ increases. Furthermore, the time variation for algorithms $SG_2, SG_3$, $RGS_1$ and $RGS_2$ are alike regardless of the changes in the number of categories $n_{Cg}$. Finally, the time variation for algorithms $SG_1, SGL_1$ are alike regardless of the changes in $n_{Cg}$.

In this article, we suggested a group of algorithms that give remarkable results. We compare the results with those of (*Sarhan, Jemmali & Ben Hmida, 2021*; *Sarhan & Jemmal, 2023*). We denote Bnew for the algorithm that gives the best value of all proposed algorithms—running the best algorithm MDETA developed in (*Sarhan, Jemmali & Ben Hmida, 2021*) on the 300 used instances. The experimental results show that for 61 instances, Bnew<MDETA. This result means that Bnew participates in giving a better solution with MDETA. Furthermore, for 134 instances, we have Bnew = MDETA. On the other hand, comparing the results given by the proposed algorithms with the best algorithm $\overline{RLT}$ developed in (*Sarhan & Jemmal, 2023*), shows that for 13 instances, we have Bnew<$\overline{RLT}$. This means that Bnew participates in giving a better solution with $\overline{RLT}$. In addition, for 116 instances, we have Bnew = $\overline{RLT}$ .Note that the proposed scheme packets protection mechanism is achieved through imposing a constraint or restrictions for the transmission of packets using a smart security policy. The **multilevel** security policy assigns security levels called categorizations to packets and uses a scheduler to control their dissemination route and dissemination time *via* the two routers to minimize the chances of breaches. The Packets which belong to the same classification level are prevented from being transmitted at the same time over the two routers.

## CONCLUSIONS

Multilevel security is one of the essential features required in future and special-purpose networks. In this research, we proposed a multilevel secure network model using two machines that can transmit confidential data in a private environment or in particular circumstances like a crisis based on a security policy. The scheme uses a scheduler to securely minimize the transmitted time when disseminating the multilevel secure packets.

We proposed several heuristics for this NP-Hard problem. The experimental results show promising results for the future development of our paradigm. Both $RGS_1$ and $RGS_4$ showed promising results. We plan for future work to increase the number of routing machines to n machines and modify our paradigm as network as a service (NaaS). We also plan for future work to demonstrate our approach in the application layer using a private set intersection and agent-based solutions to provide anonymous interaction during crisis management.

### Funding
This research was funded by the University of Jeddah, Jeddah, Saudi Arabia, under grant No. (UJ-20-DR-98). The funders had no role in study design, data collection and analysis, decision to publish, or preparation of the manuscript.

### Grant Disclosures
The following grant information was disclosed by the author:
University of Jeddah, Jeddah, Saudi Arabia: UJ-20-DR-98.

### Competing Interests
The author declares that they have no competing interests.

### Author Contributions
- Akram Y. Sarhan conceived and designed the experiments, performed the experiments, analyzed the data, performed the computation work, prepared figures and/or tables, authored or reviewed drafts of the article, and approved the final draft.

### Data Availability
   The raw data and code are available in the Supplemental Files.

### Supplemental Information
Supplemental information for this article can be found online at http://dx.doi.org/10.7717/peerj-cs.1367#supplemental-information.

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
