# Peer review of "A novel smart multilevel security approach for secure data outsourcing in crisis"

_PeerJ Computer Science, doi:10.7717/peerj-cs.1367_

## Round 0.1 · original submission · Major Revisions

Please provide the answer for each comment constructively including the explicit statement in the revised paper.

Reviewer 1 ·

Basic reporting

no comment

Experimental design

no comment

Validity of the findings

no comment

Reviewer 2 ·

Basic reporting

This article basicly meets the standards of PeerJ on language, lierature references, structure, figures, tables, and raw data shared. However, there are several minor mistakes in this manuscript and authors should proofread the manuscript carefully:
- 'Ths' -> 'This', line 195.
- 'use' -> 'uses', line 207.
- 'foes' -> 'goes', line 306.
- What's the difference between the notations 'packet Pt_j' and 'packet j' in line 227 and 228.
- The title of Figure 1 is not self-explanatory enough for readers to understand, which entitled 'Novel architecture with two routers'. What architecture did you construct? and for what?

Experimental design

- Authors enhance the previous work of (Sarhan, Jemmali & Ben Hmida, 2021), so it's better to discuss the difference and improvement of the proposed method compared with the previous work in detail.
- The introduction of the five proposed algorithms needs more detail. I suggest that the authors improve the description at lines 281-287 to provide more justification for your study. For example, why do you design these five algorithms? what is the purpose of each algorithm?
- The experimental results illustrate the effectiveness of the proposed method to minimize the transmitted time, but it seems that the authors did not discuss about how to ensure the security of data transmission using the proposed model.

Validity of the findings

The discussion of the experimental results is inadequate. The authors only show the observed results, but did not effectively summarize the results and analyze the possible reasons for the results.

Additional comments

no comment

Reviewer 3 ·

Basic reporting

The paper has poor readability due to grammatical mistakes and improper formatting of content.

The related work and introduction sections lacking focus. The author makes quite bold statements, such as the opening sentences in the abstract where is claimed that “features such as connectivity, transparency, hierarchy and openness… these featured are no longer considered significant”. I would strongly disagree, although I acknowledge many of the drawbacks of the existing Internet protocol.
Title of the paper mentions “… secure outsourcing in crisis”. However, this is not a topic discussed in the paper.

Experimental design

The author claims to propose a future network paradigm, but only offers a few experiments on average gap and transmissions time. The author states in the conclusion that the proposed paradigm shows promising results for future development. However, the presented results cover few features which the author have pinpointed as sever drawback / problems facing the internet today.

The tables lack explanation and units. The definition of the performance metrics is not very clear.

Validity of the findings

The author claims to propose a future network paradigm, but only offers a few experiments on average gap and transmissions time. R
The authors to not discuss potential drawback of the proposed paradigm.

---

## Round 0.2 · Major Revisions

As pointed out by the reviewer, please provide the answer with revision in each point explicitly to fasten the review process.

Reviewer 2 ·

Basic reporting

There are still several mistakes in the revised manuscript, e.g. "Next, Next, we describe ..." in line 304. Try to proofread the manuscript again to make it more readable.

Experimental design

The authors have already well explain the problems I raised.

Validity of the findings

Actually, the modification of the authors still does not meet my requirement. The authors just describe the data again in tables by words, but did not discover the deeper reasons hidden behind the results. For example:
- As the authors mentioned, in Table 2, "the best algorithm is the RGS" in line 395, why the RGS is better than others? ? And why the STFL is the worst?
- In Table 3, "algorithm SG2 has the lowest average gap variation of 0.01 in two scenarios when n=45 and n=55" in line 402. In total, what is the effect of n on the algorithm? Does this result indicate that the larger n is, the better the performance of the algorithm? If not, why is this happening?
- In Table 4, "all algorithms have the same average gap variation of 0.01 except algorithm SGL2" in line 405, why the algorithm SGL2 is different from others?
- In Table 5, "the algorithm SGL2 do not get impacted by the changes in the number of packets n" in line 408, why SGL2 is so special compare with others? What is the reason for this?
and so on.
I suggest that the authors could summarize each table in one or two sentences to draw some general conclusions, rather than just describing the tables again in words.

---

## Round 0.3 · accepted · Accept

Based on the positive recommendation from the reviewers, the paper is up to the standard of PeerJ. Thank you for your submission.

Reviewer 2 ·

Basic reporting

no comment

Experimental design

no comment

Validity of the findings

The authors revised the manuscript well to answer my questions.

Additional comments

Thank you for your careful revision according to my suggestions. However, please take care about the format of the manuscript, since there are several obvious minor issues, for example,
'havethe' and 'sameaverage' in line 413.

Reviewer 4 ·

Basic reporting

Nothing

Experimental design

The authors addressed all the comments and concerns, the manuscript stands for acceptance.

Validity of the findings

The authors addressed all the comments and concerns, the manuscript stands for acceptance.

Additional comments

The authors addressed all the comments and concerns, the manuscript stands for acceptance.